# The ribosome modulates folding inside the ribosomal exit tunnel

Florian Wruck[1,2,3], Pengfei Tian[4], Renuka Kudva [5], Robert B. Best [6], Gunnar von Heijne[5,7], Sander J. Tans[1,2,3 ✉] & Alexandros Katranidis [8 ✉]

Proteins commonly fold co-translationally at the ribosome, while the nascent chain emerges from the ribosomal exit tunnel. Protein domains that are sufficiently small can even fold while still located inside the tunnel. However, the effect of the tunnel on the folding dynamics of these domains is not well understood. Here, we combine optical tweezers with single-molecule FRET and molecular dynamics simulations to investigate folding of the small zinc-finger domain ADR1a inside and at the vestibule of the ribosomal tunnel. The tunnel is found to accelerate folding and stabilize the folded state, reminiscent of the effects of chaperonins. However, a simple mechanism involving stabilization by confinement does not explain the results. Instead, it appears that electrostatic interactions between the protein and ribosome contribute to the observed folding acceleration and stabilization of ADR1a.

[1] AMOLF, Amsterdam, The Netherlands. [2] Department of Bionanoscience, Delft University of Technology, Van der Maasweg 9, Delft, The Netherlands. [3] Kavli Institute of Nanoscience, Delft, The Netherlands. [4] Protein Engineering, Novozymes A/S, Lyngby, Denmark. [5] Department of Biochemistry and Biophysics, Stockholm University, Stockholm, Sweden. [6] Laboratory of Chemical Physics, National Institute of Diabetes and Digestive and Kidney Diseases, National Institutes of Health (NIH), Bethesda, MD, USA. [7] Science for Life Laboratory, Stockholm University, Solna, Sweden. [8] Institute of Biological Information Processing IBI-6, Forschungszentrum Jülich (FZJ), Jülich, Germany. ✉email: tans@amolf.nl; a.katranidis@fz-juelich.de

Protein function depends on the correct folding of the polypeptide chain into a three-dimensional structure, with the exception of disordered proteins. Folding of such structured proteins often occurs cotranslationally, i.e., at the ribosome, as the continuously elongating nascent chain emerges from the ribosomal exit tunnel[1,2]. While the majority of published studies on cotranslational folding have focused on folding outside of the ribosome[3–8], there has been an increasing interest in studying folding events within the ribosomal tunnel. Short polypeptides have been shown to assume α-helical structures in regions close to the peptidyl transferase center and near the mouth of the ribosomal tunnel[9–12], and a variety of tertiary structures, including hairpins of transmembrane helices and also domains could fit in the wider vestibule near the tunnel exit[13–15]. A variety of domains were also shown to be able to fold in the exit tunnel at linker lengths that leave part of the domain inside the tunnel[16]. In addition, cryo-electron microscopy (cryo-EM) structures on stalled translating ribosomes have provided evidence for the formation of secondary structures within the ribosomal tunnel[17]. Recent studies also suggested that small protein domains may fold inside the ribosomal tunnel[18,19].

These findings support the hypothesis that the ribosomal tunnel may not just be a passive conduit for the traversal of nascent polypeptides. Rather, it appears to play an active role in regulating the translation rate by introducing pauses and even transiently arresting protein synthesis. Variations in the translation rate can affect the folding efficiency, with slower synthesis rates reducing the probability of misfolding, allowing additional time for native-like vectorial folding to occur, particularly for complex folds[20].

However, contrary to folding outside of the tunnel and interactions of nascent polypeptides with the external surface of the ribosome[5,21], protein folding inside the tunnel has not been observed directly. The main reason for this is a lack of experimental tools to study folding within the interior of the ribosome. Single-molecule techniques have proven essential for the study of protein synthesis and subsequent folding transitions, since these highly dynamic and stochastic processes are extremely difficult to synchronize and hence observe using ensemble methods. Optical tweezers have been used to follow translation and cotranslational protein folding outside of the ribosomal tunnel in real time[8].

Here, we study the folding and unfolding of the small zinc-finger domain ADR1a both inside and outside of the ribosome to investigate the role of the tunnel during protein folding. We combine confocal fluorescence with optical tweezers, allowing correlated high-resolution force spectroscopy and single-molecule fluorescence measurements. In addition, we use molecular dynamics (MD) simulations to study the effect of the ribosomal tunnel on the folding of ADR1a.

## Results and discussion

**Experimental setup**. To study protein folding inside the ribosomal tunnel we chose the small protein domain ADR1a, a 29-residue long zinc-finger domain from yeast that folds around a $Zn^{2+}$ ion with a folding nucleus formed by two histidine and two cysteine residues[22] (Fig. 1a). A combination of experiments and simulations indicate that ADR1a can fold within the confines of the ribosomal tunnel[15,16,18], though this folding and its dynamics have not yet been observed directly.

Given the limited access within the ribosomal tunnel for post-translational attachment of fluorophores, we aimed to incorporate two fluorophores cotranslationally, at the N- and C-termini of ADR1a. We have previously shown cotranslational incorporation of two fluorophores, namely Atto633 using an amber stop codon and BODIPY-FL using the sense cysteine codon[23]. However,

since many proteins contain essential cysteines, this method is only suitable for constructs lacking them. The two cysteines of ADR1a for instance are essential for the chelation of the $Zn^{2+}$ ion, therefore we opted to make use of a 4-base codon here to incorporate TAMRA at the N-terminus of ADR1a, a fluorophore much more suitable for smFRET studies than BODIPY-FL. The acceptor dye Atto633 was cotranslationally incorporated with an amber stop codon at the C-terminus of ADR1a. Also, in order to tether the nascent chain in the optical tweezers a biotin tag was similarly introduced using another amber stop codon at the N-terminus of the nascent chain, which was synthesized by ribosomes biotinylated at the uL4 ribosomal protein[24]. After synthesis, ADR1a remained attached to the ribosome due to translational arrest at the C-terminal SecMstr arrest peptide (AP), a strong AP designed based on an extensive mutagenesis screen[25,26] (Fig. 1b and Supplementary Fig. S1). The length of the C-terminal linker was varied ($L = 26$ or $L = 34$ residues) to yield a short or long-stalled nascent chain, respectively, which positioned ADR1a either inside or outside the ribosomal tunnel.

The ribosome-nascent-chain complex (RNC) featured two biotin tags, one at the end of the stalled nascent chain and the other linked to the uL4 ribosomal protein, and was tethered between two optically trapped polystyrene beads via two identical neutravidin-DNA handles in a microfluidic chamber. Both beads were held in orthogonally polarized optical traps of equal stiffness[8]. The optical tweezers setup featured two-color confocal scanning fluorescence with single-photon sensitivity, enabling single-molecule fluorescence measurements on the incorporated fluorophores (Fig. 1c). In the folded state, the distance between the two termini of ADR1a is around 20 Å, resulting in an energy transfer between the two fluorophores with a Förster radius $R_0 = 65$ Å (Supplementary Fig. S2), and hence a fluorescence quenching of the TAMRA donor dye. A complete unfolding of the protein under tension would result in a distance between the fluorophores close to the length of the 29 residues of ADR1a in an extended conformation i.e., at least 100 Å. Hence, almost no energy transfer would occur (Supplementary Fig. S2), resulting in fluorescent emission of the TAMRA donor.

**Folding inside and outside the ribosomal tunnel**. Within the microfluidic cell of the optical tweezers, we increased the distance between the trapped beads (termed extension), first for the short ($L = 26$ residues) construct, where ADR1a is positioned inside the ribosomal tunnel, and measured the resulting force. These force ramps corresponded to the theoretically expected elastic spring-like stretching behavior of the DNA-RNC-DNA tether, as computed using the extensible worm-like chain (eWLC) model for DNA[27] and the Odijk inextensible WLC model[28] for ADR1a in series (Fig. 2a and Supplementary Data 1). Consistently, the measured curve began to deviate from the eWLC model at higher forces above 35 pN due to twisting and stretching of the rotationally unconstrained DNA linkers as they transitioned to the overstretching plateau. When zooming into the stretching data, a sudden increase in the extension was observed, suggesting unfolding of the protein domain (Fig. 2b and Supplementary Data 1). After stretching, we decreased the extension again, which yielded a progressive decrease in the measured force. During this relaxation, a sudden extension decrease and jump in the force was observed, indicating ADR1a refolding, after which the data followed the initial stretching behavior again. While the total length of the translated protein including N- and C-terminal linkers is 70–80 residues, the sudden length transitions corresponded to a contour length change of 31 amino acids on average (Supplementary Fig. S3), which agrees with the 29-residue length of the ADR1a domain. To provide additional confirmation that these

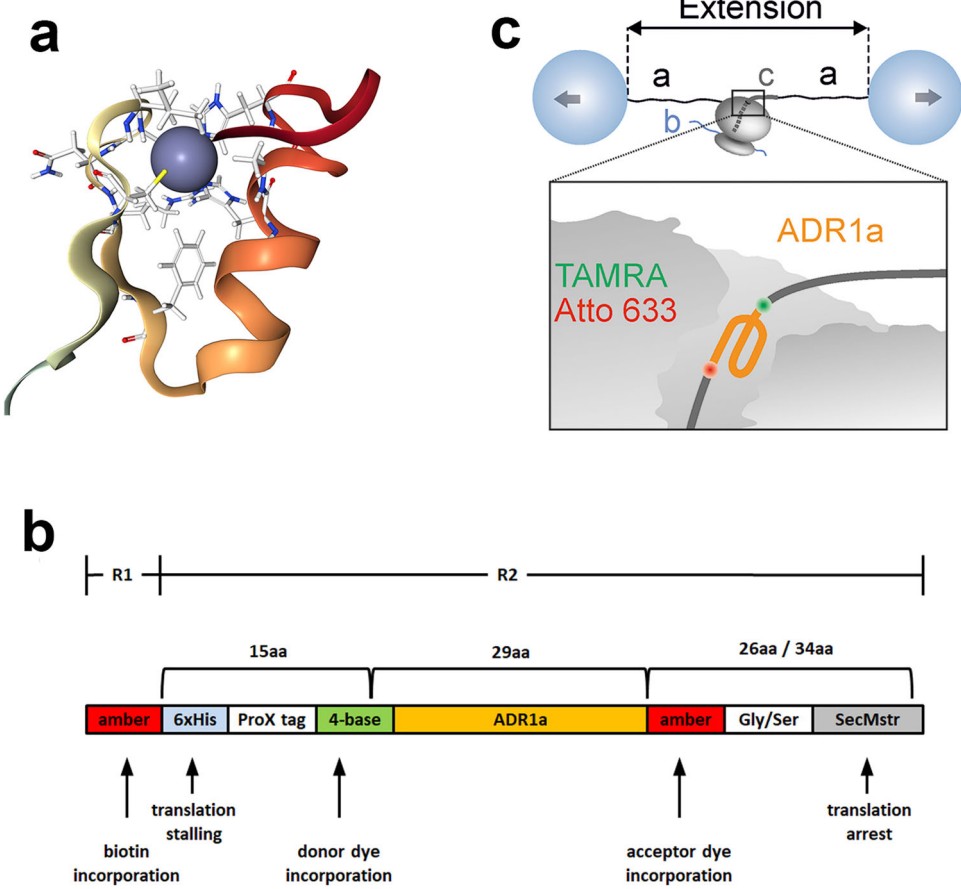

**Fig. 1 Experimental setup. a** Structure of zinc-finger protein domain ADR1a (PDB: 2ADR). The $Zn^{2+}$ ion is depicted as a blue sphere. **b** The construct used in this study. ADR1a was synthesized in a cell-free system in two reaction steps. Transcription/translation reaction 1 (R1) incorporated biotin at the N-terminal amber stop codon with protein synthesis stalled at the His-tag. After addition of histidine, protein synthesis continued and transcription/translation reaction 2 (R2) incorporated the donor dye TAMRA and the acceptor dye Atto633 at the two termini of ADR1a. The SecMstr arrest peptide ensured that the fully synthesized protein remained bound following translation. **c** Cartoon of experimental setup at the optical tweezers, where *a* depict the DNA handles, *b* the mRNA of the stalled construct, and *c* the SecMstr-stalled nascent chain.

contour length changes were due to ADR1a unfolding we performed similar unfolding experiments on a construct lacking ADR1a, consisting of a 2x Gly/Ser linker terminating in SecMstr. Here no sudden contour length changes were observed ($n = 24$ molecules, Supplementary Fig. S4). Overall, these findings indicated that folding and unfolding of ADR1a within the ribosomal tunnel could be measured using our optical tweezers assay.

Next, we tested if it was indeed possible to perform the single-molecule fluorescence measurements simultaneously during optical tweezers manipulation, to obtain further evidence for in-tunnel ADR1a folding transitions. We again performed successive extension and relaxation cycles, while repeatedly scanning a 532 nm fluorescence beam along the molecular tether and attached beads, in order to excite the TAMRA donor. Kymographs of these scans in time, which displayed the resulting TAMRA emission, showed two wide bars resulting from the bead autofluorescence. The top bead was stationary, while the steered bottom bead revealed the approach-retract movements. Notably, a narrower fluorescence signal appeared in-between the two beads, but only at higher tension (Fig. 2c, white triangles). The signal is consistent with ADR1a (un)folding i.e., ADR1a is expected to be in the folded state at lower forces, and hence TAMRA should be quenched, while in the unfolded state at higher forces TAMRA should fluoresce. Moreover, when zooming into the optical tweezers data at the appearance and disappearance of the fluorescent signal, we found they coincided

with discrete changes in ADR1a contour length that are consistent with its (un)folding (Fig. 2c, orange traces). Overall, these data provide two independent and direct observations of ADR1a unfolding and refolding within the ribosomal tunnel.

Comparing optical tweezers measurements of the short ($L = 26$) and long ($L = 34$) constructs, we found that unfolding occurred at a force of $28.5 \pm 2.0$ pN ($n = 137$ molecules, average and standard error of the mean SEM) inside, and at $25.6 \pm 2.6$ pN ($n = 88$ molecules) outside of the ribosomal tunnel (Fig. 3a and Supplementary Data 2). Refolding occurred at $18.4 \pm 1.7$ pN ($n = 93$ molecules) inside and at $14.0 \pm 1.8$ pN ($n = 49$ molecules) outside of the tunnel (Fig. 3b and Supplementary Data 2). These results thus indicated that the force required to unfold nascent ADR1a is similar, irrespective of whether the protein is inside or outside of the ribosomal tunnel. These findings suggested that any increases in unfolding force caused by interactions with the tunnel, for instance due to steric restrictions, are small or offset by compensating effects. Refolding thus occurred at a higher force inside the tunnel than outside ($p < 0.05$, Mann–Whitney U-test, two-sample *t*-test).

As a control, we also performed experiments in the absence of $Zn^{2+}$. We now observed folding much less frequently, both inside and outside the ribosomal tunnel (Supplementary Fig. S5a), in only about 16–18% of the pulling cycles. These findings are consistent with the important role of $Zn^{2+}$ ions in the ADR1a structure (Fig. 1a). When folding did occur in the absence of $Zn^2$

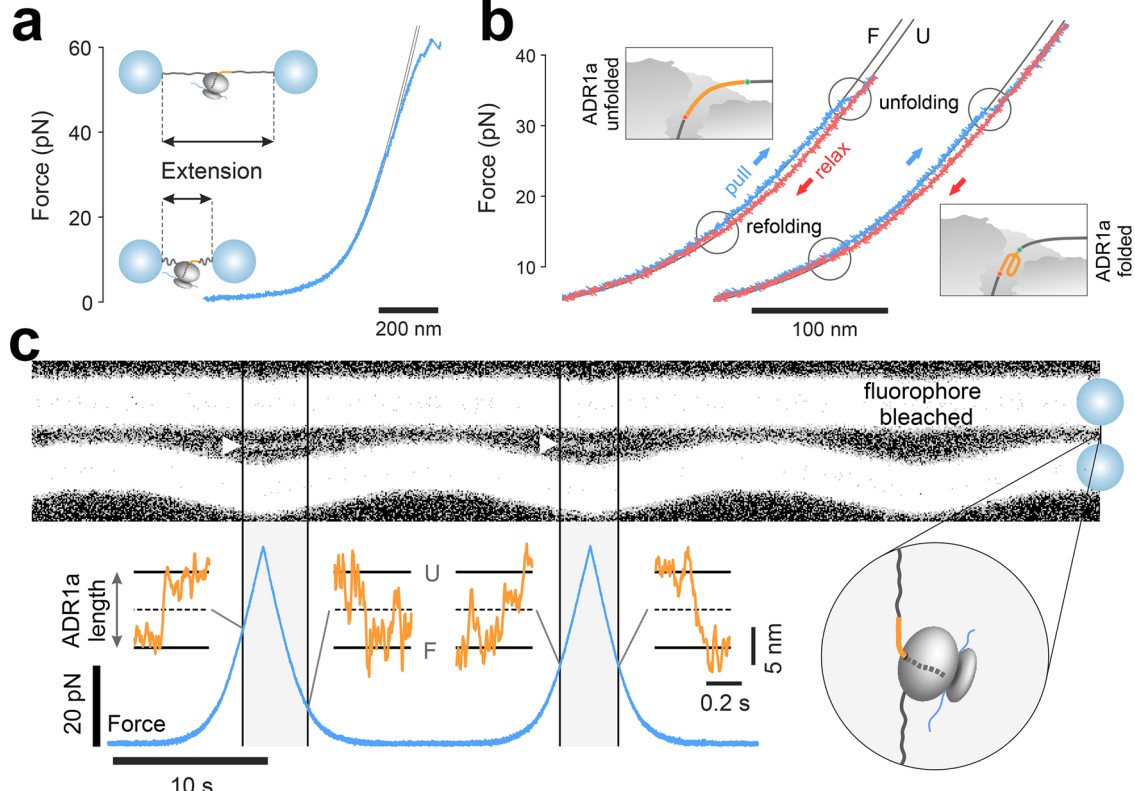

**Fig. 2 Experimental force measurements. a, b** Force-extension traces from a single ADR1a molecule undergoing unfolding and refolding transitions within the ribosomal tunnel ($L = 26$) during repeated pull-and-relax cycles, in the presence of $Zn^{2+}$. Blue curves represent pulling data, red curves relaxation data. Black curves are theoretical eWLC models representing the ADR1a folded (F) and unfolded (U) states. Cartoons show folded and unfolded ADR1a in the ribosomal tunnel. **c** Kymograph of the TAMRA donor fluorescence signal measured in-between the optically trapped beads (indicated by the white triangles), during repeated cycles of pulling and relaxing ADR1a ($L = 26$) nascent chains (top). Bottom shows corresponding force signal (blue trace). Bottom insets show the contour length of the unfolded part of ADR1a against time (orange traces), as determined using Force-Extension data, which are consistent with transitions between the unfolded (U) and folded (F) states of ADR1a.

+, the subsequent unfolding force inside the tunnel ($40.8 \pm 3.8$ pN, $n = 26$ molecules) was higher than in the presence of $Zn^{2+}$ ($28.5 \pm 2.0$ pN) or outside the tunnel in the absence of $Zn^{2+}$ ($27.1 \pm 3.4$ pN, $n = 24$ molecules) (Supplementary Fig. S5b and Supplementary Data 3). We again found a somewhat higher refolding force for ADR1a inside the ribosomal tunnel ($13.8 \pm 1.9$ pN, $n = 26$ molecules), compared to outside the tunnel ($10.2 \pm 2.1$ pN, $n = 20$ molecules), in the absence of $Zn^{2+}$ ($p < 0.05$) (Supplementary Fig. S5c and Supplementary Data 3). Note that ADR1a may adopt alternative conformations when $Zn^{2+}$ is missing, which can exhibit either lower or higher unfolding forces, as seen here in the ribosomal tunnel for instance. Also keep in mind that unfolding forces do not directly reflect unfolding barriers as measured in thermal assays, as the reaction coordinates are not the same.

These findings are notable. ADR1a folding contracts the nascent chain and hence must counteract the opposing applied force. Hence, the increased refolding forces inside the ribosomal tunnel indicate that folding occurs more readily inside than outside the tunnel, both in the presence and absence of $Zn^{2+}$. The restricted space of the ribosomal tunnel does not appear to decrease rather than increase the folding barrier. Theory suggests that confined spaces can promote folding by lowering the chain entropy, which has been proposed as a folding mechanism for the chaperonin GroEL-ES[29,30]. However, other effects may also be responsible, including (electrostatic) interactions with the tunnel surface. Nonetheless, it is notable that one cannot only directly observe the folding of a small protein domain within the

ribosomal tunnel, but that this folding is promoted, despite the steric restrictions.

**Molecular dynamics simulations**. In order to understand the role of the ribosomal tunnel in modulating the folding energy landscape of ADR1a, we used coarse-grained molecular simulations of ADR1a folding on the ribosome with different linker lengths. ADR1a and the linker with surrounding ribosomal atoms are represented by a coarse-grained model similar to our previous studies[31-34]. Each amino acid is represented by one bead at the position of the Cα atom, and each RNA base is represented by three beads at the positions of P, C4′, and N3 atoms. The interactions between the nascent chain and the ribosome are a combination of short-range repulsion representing the volume excluded by the ribosome, and electrostatic interactions given by a screened coulomb potential.

A comparison between simulations and the optical tweezers measurements was made by estimating the unfolding and folding rates of ADR1a with MD simulations at different linker lengths, corresponding to the protein being either inside ($L = 26$) or at the mouth of the ribosomal tunnel ($L = 34$) (Supplementary Fig. S6). As in the optical tweezers experiment, a force is applied between the N-terminus of the nascent chain and the N-terminus of the ribosomal protein uL4. In the simulations, a constant force is applied, and the unfolding (refolding) rate is estimated from mean first passage times computed from simulations starting in the folded (unfolded) state at each force.

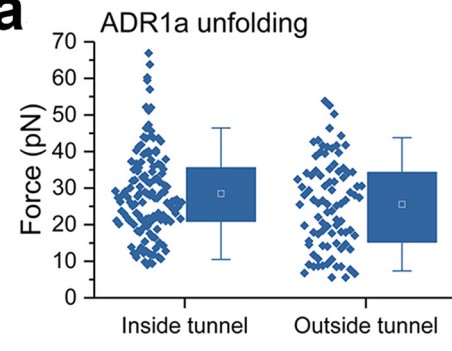

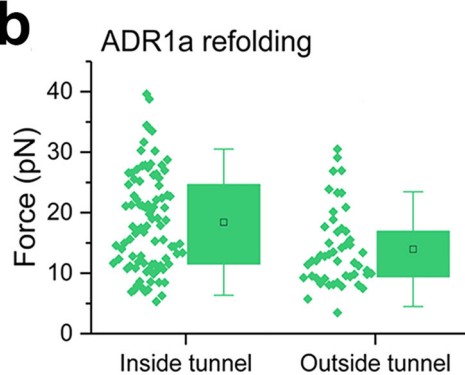

**Fig. 3 Unfolding and refolding of ADR1a with the optical tweezers in the presence of $Zn^{2+}$. a** Unfolding and **b** folding force distributions of ADR1a inside and outside of the ribosomal tunnel. The box plots show the interquartile range of the data between the 25th and 75th percentiles, the black/white squares represent the mean and the whiskers the standard deviation; the individual data points are plotted alongside.

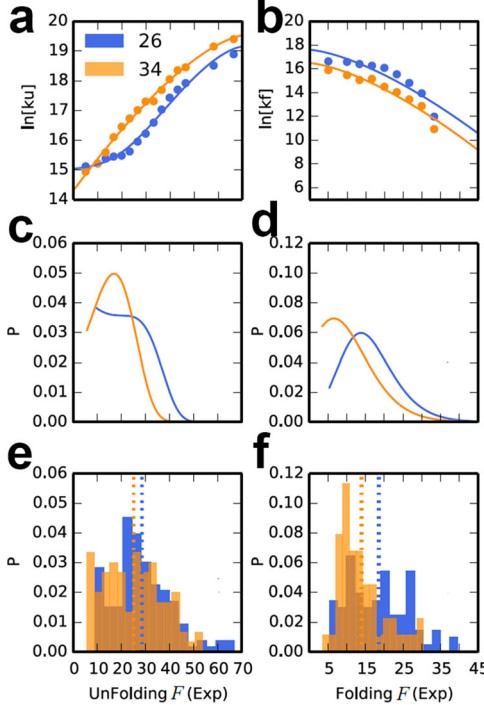

**Fig. 4 Simulated unfolding and refolding rates and force distributions of ADR1a.** Simulated force-dependent rates of ADR1a **a** unfolding and **b** folding on the ribosome. Data for $L = 26$ (ADR1a inside tunnel) and $L = 34$ (ADR1a in mouth of tunnel) are shown in blue and orange, respectively. Distributions of the corresponding **c** unfolding and **d** folding forces were computed using the force-dependent rates from simulation together with a time-dependent applied force. Corresponding experimental force distributions for unfolding and folding are shown in **e** and **f**, respectively.

As expected, the simulations showed that the application of force increased the unfolding rates and decreased the folding rates (Fig. 4a, b and Supplementary Data 4). The simulations also suggest that at $L = 26$, ADR1a unfolds slower and refolds faster than at $L = 34$, qualitatively consistent with the experimental observations. To achieve a more direct comparison with the experimental data, we calculated the theoretical unfolding (refolding) force distributions $p(F)$[35] based on the force-dependent rates $k(F)$ obtained from our simulations, together with a time-dependent change in the applied force $\dot{F}$:

$$p(F) = \frac{k(F)}{\dot{F}} e^{-\int_0^F [k(F')/\dot{F}]dF'} \tag{1}$$

The unfolding and refolding force distributions at $L = 26$ both shift to higher force compared with $L = 34$ (Fig. 4c, d and Supplementary Data 4), which is consistent with the experimental observations (Fig. 4e, f and Supplementary Data 4).

In contrast, if the electrostatic interactions between ADR1a and ribosome are not included, i.e., if there are only repulsive interactions between ADR1a and ribosome, the unfolding rate of ADR1a was very similar for $L = 26$ and $L = 34$ and the folding rate of ADR1a at $L = 26$ was in fact slower than $L = 34$ (Supplementary Fig. S7 and Supplementary Data 5), which is opposite to the trend of the experimental data. Small variations of the excluded volume parameters in the model, within a physically reasonable range, did not change this outcome. This indicates that excluded volume confinement effects often used to model such scenarios are insufficient to explain the results, perhaps because even a small applied force results in an extended unfolded state, reducing the cost of confinement. We also

systematically studied the effect of weak non-specific attractive interactions between the nascent chain and ribosome. We found that the folding rate of ADR1a of $L = 26$ remains lower than $L = 34$ (Supplementary Fig. S8 and Supplementary Data 6) at all pulling forces, which is not consistent with the ranking implied by the experimental data. Thus, the origin of the faster folding in the tunnel appears to lie in attractive electrostatic interactions between the protein domain and the tunnel wall.

A better understanding of the thermodynamics of the folding process of ADR1a was achieved by obtaining the free energy profiles of ADR1a when it folds on the ribosome with different linker lengths (Fig. 5a and Supplementary Data 7), using umbrella sampling applied along the reaction coordinate Q (fraction of native contacts)[31]. If only repulsive interactions are considered, a free energy barrier has to be overcome for both $L = 26$ and $L = 34$ in order for the protein to fold. The barrier is higher for $L = 26$, consistent with its slower folding with the model lacking electrostatics. However, addition of electrostatic interactions clearly lowers the free energy barrier in both cases, although even more pronounced for $L = 26$, rendering the transition from the unfolded to the folded state inside the tunnel much easier. For the case of $L = 40$, ADR1a is far away from the tunnel exit so that the free energy surface is identical to free ADR1a (Fig. 5a).

To obtain more insight into the lowering of the folding barrier and the stabilization of the protein when $L = 26$ in comparison to $L = 34$, free energy surfaces were determined as a function of temperature. The enthalpy and entropy contributions were calculated by fitting the stability changes at different temperatures (Fig. 5b and Supplementary Data 7), using the equation $\triangle G = \triangle E - T\triangle S$, where $\triangle E$ and $\triangle S$ represent the energy and

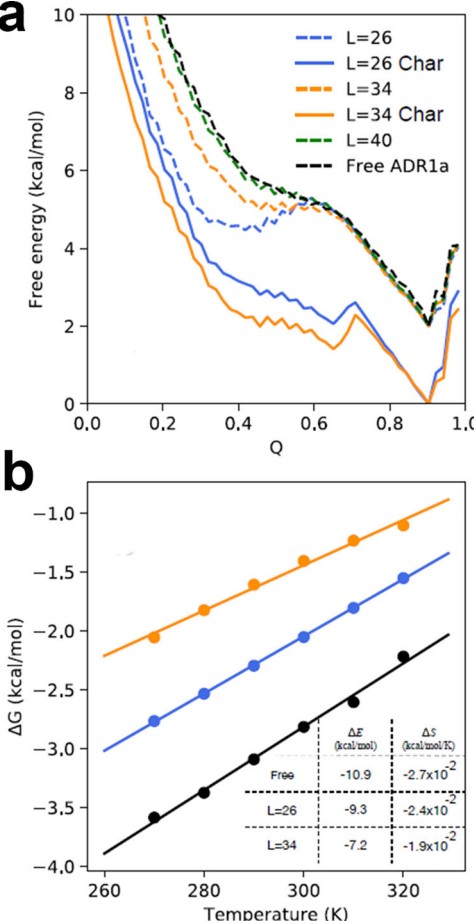

**Fig. 5 Effect of ribosome on ADR1a free energy landscape. a** Free energy profiles. Solid and dashed lines indicate results obtained with and without the inclusion of electrostatic interactions, respectively. The dashed curves are shifted vertically for visual clarity. **b** Temperature-dependent stability changes for ADR1a folding off (black) and on the ribosome with linker lengths of 26 (blue) and 34 (orange) amino acids.

entropy difference between the folded and unfolded states, respectively, and $T$ is the temperature. Note that we do not expect the contribution of $\triangle E$ and $\triangle S$ for our coarse-grained model to correspond to those that would be measured experimentally, since solvent effects are all effectively included in the energy term and are not temperature-dependent. However, for the coarse-grained model, such a decomposition allows us to readily separate effects coming from protein configuration entropy, which is the only contribution to $\triangle S$ in the model, from the free energy of interaction with the ribosome in $\triangle E$.

The thermodynamic analysis shows that on the ribosome, for both $L = 26$ and $L = 34$, ADR1a is entropically stabilized relative to the ribosome free case due to the confinement effect[29]. However, compared to the case of free ADR1a, the ribosome energetically destabilizes ADR1a at both $L = 26$ and $L = 34$, most likely because the unfolded state exposes more surface area and therefore binds more favorably to the ribosome than the folded state[36]. Overall, this loss of stability due to energetic effects is larger than the entropic stabilization by confinement (for both $L = 26$ and $L = 34$).

It is clear both from the experimental measurements as well as from the simulations that the ribosome modulates the folding energy landscape. Although the faster folding rate and greater stabilization of the protein when deeper in the tunnel are very

suggestive of stabilization due to a simple excluded volume confinement effect, the simulations indicate that in fact confinement alone does not explain the results. Our study suggests that electrostatic interactions with the ribosomal tunnel facilitate folding by lowering the folding free energy barrier and stabilizing the folded state similar to the proposed function of chaperonin during protein folding[36].

## Methods

**Isolation of biotinylated ribosomes**. Ribosomes from Can20/12E[37], an RNAse deficient *Escherichia coli* K-12 strain was biotinylated in vivo at the uL4 ribosomal protein and subsequently isolated as previously described[8]. The activity of the biotinylated ribosomes was checked by synthesizing GFP emerald (GFPem) in vitro (PURExpress Δ ribosomes, NEB #E3313S, New England Biolabs) and measuring fluorescence of GFPem in a QM-7 spectrofluorometer (Photon Technology International, Birmingham, NJ).

**Plasmid construction**. The gene encoding for ADR1a was inserted into the pRSET plasmid (Thermo Fisher Scientific), between the NdeI and XhoI restriction sites. The amber stop codon TAG was added upstream the gene of ADR1a, followed by a sequence encoding for 6 histidines (6×His) and the ProX tag peptide that contains the four-base codon CGGG for incorporation of unnatural amino acids[38]. Similarly, another amber stop codon TAG was engineered downstream of the ADR1a gene, followed by a Gly/Ser rich linker and an enhanced variant of the SecM arrest peptide (SecMstr) (FSTPVWIWWWPRIRGPP)[26], between the XhoI and HindIII restriction sites. The C-terminal extension was either engineered to be 26 or 34 residues long to allow folding of the protein domain inside or outside the ribosomal tunnel, respectively.

For the control experiments an amber stop codon TAG, followed by the same 34 residues long C-terminal extension (Gly/Ser rich linker and SecMstr), was inserted between the NdeI and HindIII restriction sites.

**Coupling of neutravidin-DNA handles to beads**. 2.1-μm diameter carboxyl-functionalized polystyrene beads (Spherotech) were modified with anti-digoxigenin (anti-DIG, Roche), using the carbodiimide crosslinker EDAC. Double-stranded DNA (dsDNA) molecules were prepared by PCR amplification using digoxigenin (DIG) and biotin 5′-end-modified primers. The resulting 5 kbp PCR fragments (Bio-DNA-DIG) (1.7 μm in length) reacted with neutravidin (Sigma) at a ratio of 100 neutravidin/DNA for 24 h at 4 °C and were subsequently coupled to the anti-DIG beads at a reaction ratio of 150 neutravidin-DNA/bead for 30 min at 4 °C. The beads with the neutravidin-DNA handles were then washed several times with Tico buffer (20 mM HEPES-KOH pH 7.6, 10 mM (Ac)$_2$Mg, 30 mM AcNH$_4$, 4 mM β-mercaptoethanol as an additional oxygen scavenger) and split into two batches[39].

**Coupling of ribosomes to beads with DNA handles**. Biotinylated ribosomes were mixed with one batch of Tico-washed neutravidin-DNA modified beads (>1000 ribosomes per accessible neutravidin binding pocket) and incubated at 4 °C for 30 min. Excess unbound ribosomes were removed by pelleting and the beads were washed once with Tico buffer and resuspended directly in the first cell-free transcription/translation mix described below.

**Cell-free protein synthesis and cotranslational labeling**. The cell-free transcription/translation mix used in this study is a customized version of the PURE system[40] without ribosomes (PURExpress Δ ribosomes, New England Biolabs) and the amino acid histidine. Protein synthesis was carried out in two steps.

In the first reaction step, the system was supplemented with 10 μM of a modified tRNA pre-charged with biotinylated lysine. Synthesis was initiated by mixing in the system the ribosomes coupled to beads and 5.5 nM linearized plasmid. The reaction mixture was incubated at 37 °C for 20 min.

Biotin was incorporated cotranslationally at the N-terminal amber position TAG using the suppressor tRNA technique, as described previously[41]. Since the cell-free system lacked histidine, synthesis was prematurely paused upon reaching the 6×His sequence (Fig. 1b reaction 1). Excess of the pre-charged tRNA was removed by pelleting the stalled RNC complexes coupled to beads, washing them on ice in Tico buffer, and resuspending them directly in the second cell-free reaction mix.

In the second reaction step, 150 μM of the amino acid histidine was added to the system, which was additionally supplemented with two different modified tRNAs, pre-charged with phenylalanine labeled with TAMRA or Alexa633, respectively (Clover Direct, Tokyo, Japan). Synthesis continued after adding this system to the previously stalled RNC complexes coupled to beads. The reaction mixture was incubated at 37 °C for additional 20 min. Phenylalanine labeled with TAMRA was incorporated at the N-terminus making use of the four-base codon CGGG, while phenylalanine labeled with Alexa633 was incorporated at the C-terminus of the protein via the amber stop codon TAG. The protein labeled with the incorporated FRET pair remained attached to the ribosome due to the SecMstr

arrest peptide (Fig. 1b reaction 2). Following the second transcription/translation reaction the bead-tethered stalled RNC complexes were resuspended in Tico buffer (20 mM HEPES-KOH pH 7.6, 10 mM (Ac)$_2$Mg, 30 mM AcNH$_4$, 4 mM β-mercaptoethanol), either supplemented with 50 μM Zn(Ac)$_2$ or with 50 μM TPEN in control experiments without Zn$^{2+}$. The sample was subsequently injected into the microfluidic chamber. As oxygen scavenger, a combination of 1 mM Trolox and the P2O system (3 units per ml pyranose oxidase, 90 units per ml catalase, and 50 mM glucose, Sigma) were used.

**Optical tweezers setup**. Correlated single-molecule force spectroscopy and multi-color confocal laser scanning spectroscopy measurements were carried out with the C-trap instrument (Lumicks, Amsterdam). This instrument features two high-resolution optical traps formed by a powerful intensity- and polarization-stable single 1064 nm laser, which is split into two orthogonally polarized beams. Trap stiffness was kept at $260 \pm 50$ pN μm$^{-1}$ for all measurements. Two fluorescence excitation lasers (532 and 638 nm) allow for two-color confocal fluorescence, while the dedicated APDs assure single-photon sensitivity. Measurements were performed in a monolithic laminar flow cell with a very stable passive pressure-driven microfluidic system with five separate flow channels.

**Molecular simulations**. Simulations were run with a coarse-grained potential taken from our earlier work on the folding of translationally arrested proteins[31]. The model has a single bead per protein residue and three beads per nucleic acid residue, with the bead radii chosen to approximate the excluded volume of each residue in crystal structures. In addition to the energy terms in the former model, we have added an electrostatic potential with the functional form

$V = \sum_{i<j} \frac{q_i q_j}{4\pi\varepsilon_d\varepsilon_0} e^{-\frac{d_{ij}}{\lambda_D}}$, where $q_i$ and $q_j$ are the charges on atoms $i$ and $j$, respectively, $d_{ij}$ is the distance between atoms $i$ and $j$, $\lambda_D$ is the Debye screening length, $\varepsilon_0$ is the permittivity of free space, and $\varepsilon_d$ is a distance-dependent dielectric permittivity correction[42].

$\varepsilon_d(r) = (\frac{5.2+\varepsilon_s}{2}) + (\frac{\varepsilon_s-5.2}{2})\tanh[(\frac{r-r_{m\varepsilon}}{\sigma_\varepsilon})]$, where $\varepsilon_s$ is the dielectric constant of water (80 at 298 K) and the factor of 5.2 is the limiting dielectric in the vicinity of the charged residue. The location of the inflection/midpoint of the curve $r_{m\varepsilon}$ is 8 Å and the switching distance scale $\sigma_\varepsilon$ is 10.0 Å. The charges are $-1$ for aspartate, glutamate, and DNA nucleotide, $+1$ for arginine and lysine, and $+0.5$ for histidine.

Force-dependent rates $k(F)$ were obtained from mean first passage times for folding or unfolding, determined by running a large number of simulations with constant force, starting from unfolded or folded states, respectively.

The theoretical unfolding (refolding) force distributions $p(F)$ were calculated based on the force-dependent rates $k(F)$ obtained from our simulations, together with a time-dependent change in the applied force $\dot{F}$:

$$p(F) = \frac{k(F)}{\dot{F}} e^{-\int_0^F [k(F')/\dot{F}]dF'} \tag{1}$$

where

$$k(F) = k_0 \left(1 - \frac{vFx^\ddagger}{\triangle G^\ddagger}\right)^{\frac{1}{v}-1} e^{\triangle G^\ddagger[1-(1-vFx^\ddagger/\triangle G^\ddagger)^{\frac{1}{v}}]} \tag{2}$$

with $v = 2/3$. $k(F)$ is calculated by fitting Eq. (2) to the rates obtained by the simulations as illustrated in Fig. 4a, b.

**Statistics and reproducibility**. No statistical methods were used to predetermine sample size. Sample sizes were chosen based on previous experience and published studies to assess reproducibility. Experiments were replicated multiple times on multiple substrate molecules using different bead pairs, which were sufficient to obtain the described statistical significance. All attempts at replication were successful. Sample sizes $N$ are mentioned in the main text for each measurement.

**Reporting summary**. Further information on research design is available in the Nature Research Reporting Summary linked to this article.

## Data availability
Data that support the findings of this study are available from the corresponding author upon reasonable request. Original data to produce plots shown in Figs. 3, 4, and 5, as well as Supplementary Figs. S5, S7, and S8 are available as Supplementary Data.

## Code availability
All custom-made Matlab and python scripts used in data analysis are available upon request. The custom code can be downloaded via the link https://bitbucket.org/marioavellaneda/foldometer/downloads/.

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

## Acknowledgements

F.W. received funding from the European Union's Horizon 2020 Research and Innovation Program under the Marie Skłodowska-Curie grant agreement No. 745798. R.B.B. was supported by the Intramural Research Program of the National Institute of Diabetes and Digestive and Kidney Diseases of the National Institutes of Health. This work utilized the computational resources of the NIH HPC Biowulf cluster (http://hpc.nih.gov). Work in the group of S.J.T. is supported by the Netherlands Organization for Scientific Research (NWO). This work was supported by grants from the Knut and Alice Wallenberg Foundation (2012.0282), the Novo Nordisk Fund (NNF18OC0032828), and the Swedish Research Council (621-2014-3713) to G.v.H. This project was supported by the Deutsche Forschungsgemeinschaft, DFG, project number KA 4388/2-1 to A.K.

## Author contributions

F.W., P.T., R.K., R.B.B., G.v.H., S.J.T., and A.K. designed the experiments. F.W. performed the optical tweezers measurements. P.T. performed the MD simulations. F.W., P.T., R.K., R.B.B., G.v.H., S.J.T., and A.K. analyzed the data and discussed the results. R.B.B., G.v.H., S.J.T., and A.K. supervised the study. A.K. coordinated the project. F.W., P.T., R.K., R.B.B., G.v.H., S.J.T., and A.K. wrote and edited the manuscript.

## Funding

## Competing interests

The authors declare no competing interests.
