## [Peer Review File · Communications Biology]

Reviewers' comments:

Reviewer #1 (Remarks to the Author):

Review of COMMSBIO-20-2211-T

The problem addressed in this manuscript is of general interest. However, it is unclear how much the work it describes has advanced our understanding of it.

1. There are two control experiments that appear not to have been done. (a) The protein of concern binds Zn^{2+} . All of the tweezer experiments were done in the presence of that ion. The authors should have done experiments in its absence. This would change the energetics of the folding/unfolding reaction, and if the forces at which unfolding were observed to drop in Zn-free buffers, it would support the claim that the discontinuity in the force/length traces seen represent the unfolding of that protein. (b) Similar experiments should have been done using a peptide that was all-linker. In that case, there should have been no such discontinuity.

2. The attachment point for the linker between the bead and the ribosome is uL4. uL4 is adjacent to the distal end of the exit tunnel. This means that the forces applied were close to perpendicular to the axis of the tunnel rather than parallel to it. The distal end of the tunnel was being pulled open in these experiments. This raises the possibility that the force/length discontinuity observed represents some kind of reversible failure of the structure of the ribosome rather than denaturation/renaturation of the protein of concern.

3. The molecular weight of stuff the authors had to add to the very small protein they were intent on studying in order to work with it vastly exceeds the molecular weight of that protein. It includes two organic dye molecules that are highly hydrophobic. What assurance do we have that this construct folds and unfolds the same way as the small protein of concern?

4. The computer simulations done took only electrostatic interactions into account. van der Waals and hydrogen bonding interactions were ignored. Why is that? In addition, how can anyone be sure that the authors' have correctly evaluated the electrical potential distribution inside the tunnel. The tunnel is surrounded by charged groups. The structure of the counterion atmosphere is poorly understood. It does not sound likely.

Reviewer #2 (Remarks to the Author):

In this manuscript, the authors use a number of techniques applied to ADR1 zinc-finger protein to find that the folding rate is faster inside the exit tunnel than outside, and that the midpoint unfolding and folding forces are larger inside than outside the exit tunnel. The authors provide simulation evidence that it is not the confining effect of the tunnel that causes this, rather it is electrostatic interactions with the exit tunnel. This is a nice contribution to the field, illustrating that the exit tunnel can enhance folding.

Major concerns:

1. In the introduction, the author's don't characterize the state of the field accurately regarding previous simulation work on protein folding in the exit tunnel. They state that hairpins have been shown to fold in the tunnel and cite a simulation paper, however, that study showed a variety of tertiary structure (including domains) could fit in the exit tunnel (PMID: 21062068), and another simulation study showed a variety of domains can fold in the exit tunnel (PMID: 21204555) at linker lengths that leave part of the domain inside the tunnel. I think the intro should be updated to clearly state this, and cite the additional paper.

2. A statistical test must be applied to the key experimental result in Fig. 3 to test if the differences between the distributions are statistically significant. For example, the Mann-Whitney U-test, or a random permutation test would be acceptable ways to compute the p-value of the

observed difference. Given there 95% CI's are small, I expect the difference to be significant.

3. The authors conclude electrostatics are important based simulation results. Why did they not change the counter-ion concentration in the experiment to test this prediction?

Reviewer #1 (Remarks to the Author):

Review of COMMSBIO-20-2211-T

The problem addressed in this manuscript is of general interest. However, it is unclear how much the work it describes has advanced our understanding of it.

1. There are two control experiments that appear not to have been done. (a) The protein of concern binds Zn²⁺. All of the tweezer experiments were done in the presence of that ion. The authors should have done experiments in its absence. This would change the energetics of the folding/unfolding reaction, and if the forces at which unfolding were observed to drop in Zn-free buffers, it would support the claim that the discontinuity in the force/length traces seen represent the unfolding of that protein. (b) Similar experiments should have been done using a peptide that was all-linker. In that case, there should have been no such discontinuity.

Author Response: Indeed, these control experiments are important and we thank the reviewer for pointing this out. We performed two sets of control experiments: (a) both ADR1a constructs in the absence of Zn²⁺, carried out with 50 μM of the Zn²⁺ chelator TPEN in all buffers, and (b) using a peptide that was all-linker (pRSET + amber + 2x Gly/Ser + SecMstr). In control experiment (b) we add a repeat of the original Gly/Ser linker of the ADR1a construct L= 34 to ensure that the construct spans the entire length of the ribosomal tunnel so that the N-terminal biotin can still be tethered *in situ*.

In the absence of Zn²⁺ (a) we find that ADR1a folds only for 18.3 % of all pulling cycles inside the ribosomal tunnel and 16.7 % of pulling cycles outside the tunnel (new Fig. S5A). In contrast, in the presence of Zn²⁺ ADR1a folding was observed for 78.3 % of all pulling cycles inside and 73.9 % of all pulling cycles outside the ribosomal tunnel. Hence, ADR1a folding is significantly reduced in the absence of Zn²⁺, which supports our assertion that the discontinuity indeed represents ADR1a unfolding.

When folding did occur in the absence of Zn²⁺, the subsequent average unfolding force inside the ribosomal tunnel is higher (40.8 ± 3.8 pN, N = 26, average and standard error of the mean SEM) than in the presence of Zn²⁺, or outside the tunnel in the absence of Zn²⁺ (27.1 ± 3.4 pN, N = 24) (new Fig. S5B). Perhaps, ADR1a can adopt non-native conformations in the absence of Zn²⁺, giving rise to higher unfolding forces of such misfolded states. In the absence of Zn²⁺, we do again find somewhat higher refolding forces for ADR1a inside the ribosomal tunnel (13.8 ± 1.9 pN, N = 26), compared to outside the tunnel (10.2 ± 2.1 pN, N = 20) (new Fig. S5C). An additional paragraph is also added now in the text (p.7, lines 157-166).

In control experiment (b), which is all-linker, we consistently do not find any unfolding of the nascent chain, as shown in the new Fig. S4. The main text is also updated (p.5, lines 117-121)

Both control experiments support the claim that the force discontinuities represent ADR1 unfolding.

2. The attachment point for the linker between the bead and the ribosome is uL4. uL4 is adjacent to the distal end of the exit tunnel. This means that the forces applied were close to perpendicular to the axis of the tunnel rather than parallel to it. The distal end of the tunnel was being pulled open in these experiments. This raises the possibility that the force/length discontinuity observed represents some kind of reversible failure of the structure of the ribosome rather than denaturation/renaturation of the protein of concern.

Author Response: The reason we use uL4 is that it is one of the few ribosomal proteins that penetrate the rRNA instead of just sitting on its surface. This ensures that by applying a force we do not pull the protein out. We used the same approach successfully in a previous work (Wruck et al. PNAS 2017) studying the synthesis of three proteins that folded under a different force range. The observed force-extension traces agreed with the expected unfolding/folding force range, giving no indication of a reversible structural failure of the ribosome. Moreover, the additional controls further confirm that the contour length increases reflect ADR1a unfolding.

3. The molecular weight of stuff the authors had to add to the very small protein they were intent on studying in order to work with it vastly exceeds the molecular weight of that protein. It includes two organic dye molecules that are highly hydrophobic. What assurance do we have that this construct folds and unfolds the same way as the small protein of concern?

Author Response: Regarding these questions on the organic dyes, we first stress that our conclusions do not concern absolute measurements of quantities, but rather are based on how these quantities change depending on conditions and construct design, while keeping all other experimental aspects including DNA tethering and dye labeling the same. We further note that the labeling of proteins with dye molecules is a standard approach in fluorescence / FRET measurements that has been followed in numerous studies, including those where dyes are incorporated co-translationally (Holtkamp et al. Science 2015 / Coi et al. ACS Synth.Biol. 2017 / Sadoine et al. Anal.Chem. 2018 are just a few of them). In all these studies hydrophobic linkers and the dyes themselves do not seem to interfere with the folding, and are established as a valid approach.

4. The computer simulations done took only electrostatic interactions into account. van der Waals and hydrogen bonding interactions were ignored. Why is that? In addition, how can anyone be sure that the authors' have correctly evaluated the electrical potential distribution inside the tunnel. The tunnel is surrounded by charged groups. The structure of the counterion atmosphere is poorly understood. It does not sound likely.

Author Response: In the spirit of coarse-grained modeling we used only the minimal interactions necessary to describe the system. We started with the excluded volume effect of the ribosome because that is essential to include and can explain the folding of many proteins in coarse-grained simulations by our group and others (e.g. Ed O'Brien, Tom Miller). We then added a treatment of screened electrostatics via Debye-Huckel theory as a first approximation to the interaction of the protein with the highly charged ribosome tunnel. We found that these two effects in combination were sufficient to explain the data. Non-electrostatic interactions may be important too, but adding a generic short-range attraction was not enough to explain the difference -- only the electrostatics reproduced the order of unfolding/folding forces observed in experiment. Hence, we believe that electrostatics are the main factor. We have now added the simulations with non-specific interactions to the SI (Fig. S8) and we discuss them in the main text (p.10, lines 225-228). We find that using only non-specific interactions between ribosome and nascent chain does not give the correct order of folding rate ($k_{\text{fold_L26}} > k_{\text{fold_L34}}$). We thank the referee for raising this issue, which led us to revise the manuscript to clarify our reasoning behind the model used.

Reviewer #2 (Remarks to the Author):

In this manuscript, the authors use a number of techniques applied to ADR1 zinc-finger protein to find that the folding rate is faster inside the exit tunnel than outside, and that the midpoint unfolding and folding forces are larger inside than outside the exit tunnel. The authors provide simulation evidence that it is not the confining effect of the tunnel that causes this, rather it is electrostatic interactions with the exit tunnel. This is a nice contribution to the field, illustrating that the exit tunnel can enhance folding.

Major concerns:

1. In the introduction, the author's don't characterize the state of the field accurately regarding previous simulation work on protein folding in the exit tunnel. They state that hairpins have been shown to fold in the tunnel and cite a simulation paper, however, that study showed a variety of tertiary structure (including domains) could fit in the exit tunnel (PMID: 21062068), and another simulation study showed a variety of domains can fold in the exit tunnel (PMID: 21204555) at linker lengths that leave part of the domain inside the tunnel. I think the intro should be updated to clearly state this, and cite the additional paper.

Author Response: We thank the reviewer for the comments and suggestions on the introduction. The text and the corresponding citations are now updated (p.2 lines 34-37).

2. A statistical test must be applied to the key experimental result in Fig. 3 to test if the differences between the distributions are statistically significant. For example, the Mann-Whitney U-test, or a random permutation test would be acceptable ways to compute the p-value of the observed difference. Given there 95% CI's are small, I expect the difference to be significant.

Author Response: We had performed a two-sample t-test in the original manuscript to determine if the unfolding and refolding force distributions in Fig. 3 were statistically significant. With a p-value of 0.001 we found that the difference in the mean refolding forces inside and outside the tunnel are indeed significantly different (<0.05). The difference in the mean unfolding force distributions inside and outside the tunnel are not significantly different with a p-value of 0.08. Additionally we have now also conducted a Mann-Whitney U-test (p.7 line 155-156), and find a p-value of 0.0015 for the two refolding force distributions and 0.13 for the two unfolding force distributions.

3. The authors conclude electrostatics are important based simulation results. Why did they not change the counter-ion concentration in the experiment to test this prediction?

Author Response: We thank the reviewer for this suggestion. We indeed considered changing counter-ion concentrations, however the effect is expected to be minor because of the very limited concentration ranges compatible with ribosome stability and function, and the fact that the interactions in question occur at very short length scales (direct salt bridges with the ribosome tunnel), which would be less affected by salt screening, with mono-valent cations requiring even more range than divalent cations. The divalent cations are limited to a concentration range between 5 and 15 mM for ribosomal function. The monovalent ion concentration can be varied over a range between 20 and 500 mM salt.

REVIEWERS' COMMENTS:

Reviewer #1 (Remarks to the Author):

The authors are to be commended for having run the control experiments necessary to test the hypothesis that the length changes they describe reflect the folding and unfolding of the Zn-finger domain of concern. Unfortunately, those control experiments demonstrate that the phenomenon they have been studying is more complicated than they thought. In the absence of Zn, the fraction of the experiments they did that showed the length discontinuities expected for folding and unfolding events dropped by a factor of ~ 4 , but the force required to induce length increases in those experiments where length extensions were seen increased a lot in the ribosome, while remaining at about at their + Zn values outside the ribosome. These observations are hard to explain. The two reasonable outcomes would have been: (1) no change in the fraction of experiments showing length changes, but a reduced force required for unfolding, or (2) no experiments showing any length changes at all (i.e. domain so destabilized by the absence of Zn that it does not fold).

The authors would probably be wise to withdraw this mms.

Reviewer #2 (Remarks to the Author):

The authors have addressed my concerns. I recommend publication of this version of the manuscript.